# Weighting Factor Selection of the Ensemble Model for Improving Forecast Accuracy of Photovoltaic Generating Resources

**Kihan Kim and Jin Hur *** 

Department of Energy Grid, Sangmyung University, Seoul 03016, Korea
* Correspondence: jinhur@smu.ac.kr; Tel.: +82-02-781-7576

**Abstract:** Among renewable energy sources, solar power is rapidly growing as a major power source for future power systems. However, solar power has uncertainty due to the effects of weather factors, and if the penetration rate of solar power in the future increases, it could reduce the reliability of the power system. A study of accurate solar power forecasting should be done to improve the stability of the power system operation. Using the empirical data from solar power plants in South Korea, the short-term forecasting of solar power outputs were carried out for 2016. We performed solar power forecasting with the support vector regression (SVR) model, the naïve Bayes classifier (NBC), and the hourly regression model. We proposed the ensemble method including the selection of weighting factors for each model to improve forecasting accuracy. The forecasted solar power generation error was indicated using normalized mean absolute error (NMAE) compared to the plant capacity. For the ensemble method, the results of each forecasting model were weighted with the reciprocal of the standard deviation of the forecast error, thus improving the solar power outputs forecast accuracy.

**Keywords:** ensemble; support vector regression; naïve Bayes classifier; machine leaning; day ahead power forecasting; solar power forecasting

## 1. Introduction

With the recent launch of a new climate system around the world, the composition of the entire country is changing. In 2017, the new renewable resources capacity was 178 GW, with solar power being the top power generation source among new power generation types [1]. According to global trends, Korea is also planning to introduce large-scale, variable power systems through the renewable energy 3020 policy. According to the 8th Basic Plan for Electricity Supply, it is planned to accommodate a large-scale (58.5 GW) of renewable resources to supply 20% of the required power generation. Of the total, solar power is equivalent to 36.5 GW, or 62 percent, of the supply of renewable energy. Figure 1 shows the current state of accumulated solar power facilities in South Korea as of 2017 [2]. Solar power generation is concentrated in Jeollanam-do Province and Jeollabuk-do Province, where solar radiation conditions are good.

Unlike conventional power sources, solar power has the characteristic of varying output due to various meteorological causes such as solar radiation, temperature, and cloud cover amount, which can cause instability in the system due to the variability and uncertainty of solar power generation when large solar power complexes are connected to the system [3]. Measures need to be taken to improve the flexibility of the system in response to the increase in the system linkage of variable power. Among the many measures to ensure system flexibility, accurate forecasting of renewable energy output is considered a cost-effective method [4]. It was also found that the improvement of

the forecasted accuracy of the renewable energy will result in significant economic benefits as well as improve the reliability of the system, including reduction of the operating costs of the power system.

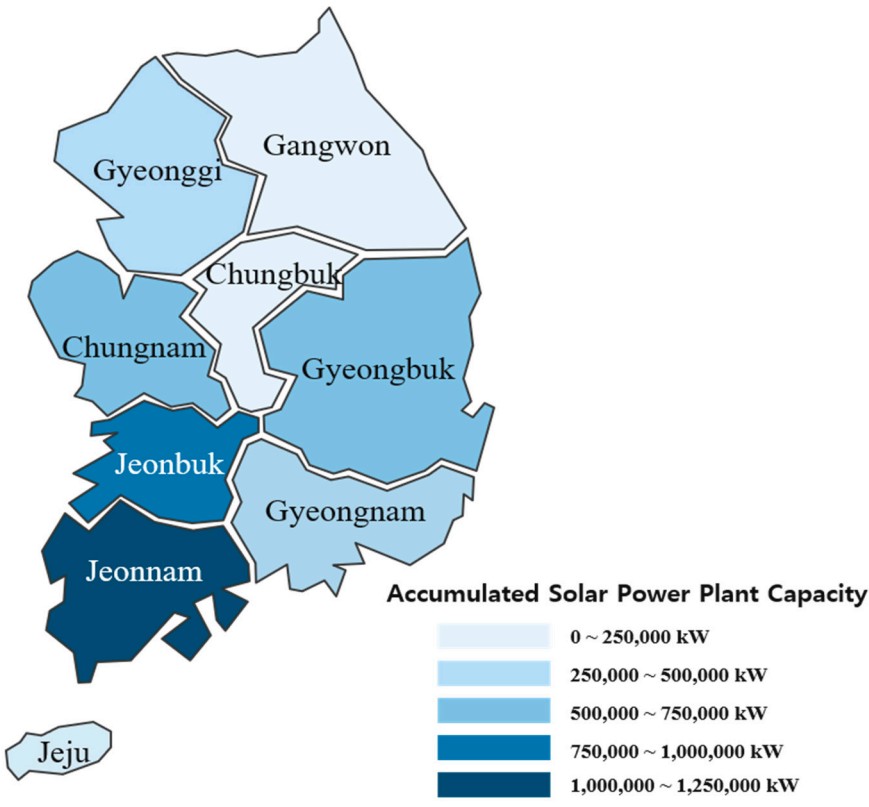

**Figure 1.** Korea's accumulated capacity of photovoltaic power plants in 2017.

In sum, forecasting solar power output is necessary for increasing the supply level of the power system and for the efficient operation of power systems. Various short-term photovoltaic forecasting methods have recently been studied. The persistence method assumes that the forecasted power for the time horizon is equal to the last value measured [5]. The photovoltaic performance method is a forecasting method using the relationship between insolation and solar output. Because no historical data is needed, the power output of the plant can be obtained before construction [6]. Statistical models do not require solar power nameplate data for modeling. It is a data-driven technique that uses the relationship to historical data based on the forecasting of solar power generation. This model includes auto-regression (AR), auto-regressive moving average (ARMA) [7,8], and forecasting photovoltaic power generation at different cycles and analyzing their characteristics over time [9]. The regressive analysis shows the relationship between the dependent variable (photovoltaic output) and various independent variables (meteorological data) and forecasts the solar power output obtained through the regression function when new weather data is inputted. The artificial neural networks (ANN) technique is a method of making solar power forecasts using machine learning methods and was inspired by the way neurons work. Through training, weights of each layer are selected, and output forecasting is performed using them [10–12]. One of the simplest machine learning methods is k-nearest neighbors (k-NN). It is a forecasting algorithm based on a pattern recognition algorithm that compares the current state with the training sample. By calculating the training data and Euclidean distance, the nearest k neighbors are selected [13]. In order to improve the forecasting model, research is being conducted to combine hourly regression models, statistical models and artificial intelligence (AI) models [14–16]. U.S. power system operators have applied forecasting models based on persistence, numerical weather prediction (NWP), and statistical methods, and have recently adopted ensembles and probabilistic forecasting models [17,18].

In this paper, the ensemble technique is presented for upgrading the forecasting of solar power output. Section 2 introduces methods for NBC, SVR, and hourly regression models. Section 3 presents the use of empirical data for forecasting of solar power for each model and results are presented using NMAE. Section 4 uses an ensemble technique to improve the accuracy of forecasting solar power by using the forecast result for a single model to achieve enhancement of solar power output forecasting.

## 2. Power Output Forecasting Model of Photovoltaic Generating Resources

### 2.1. NBC Model

The NBC forecasting model is a forecasting model that creates classification rules based on historical data and is classified according to predefined classification rules when new forecast conditions are applied [19–24]. NBC's forecasting model is effectively trained in a supervised learning environment and is often used in areas such as document classification and disease forecasting [25,26]. The NBC has the advantage of being very simple in its conception and assumption, resulting in a small amount of training data required to estimate [27,28]. Machine learning methods can struggle if too many variables are used, such as overfitting, but NBC techniques have no limitations in selecting variables because scalability is good. In addition, in this study, classification variable was selected as output data of solar power and auxiliary variables as meteorological data. The NBC model is a method of expressing the relationship between the pre- and post-probability of a probability variable, such as expression as shown in Equation (1) based on Bayes probability theorem [19–24].

$$P(A|B) = \frac{P(B|A) \times P(A)}{P(B)} \tag{1}$$

In the above Equation (1), $P(A)$ is the prior probability of event A, which means not knowing any information about event B. $P(B|A)$ is the conditional probability of event B when event A is given, and it is determined according to the classification criteria. $P(A|B)$ is the post-probability of event A for which the value of event B is given. At this time, the probability of event A changes from $P(A)$ to $P(A|B)$ after event B is observed. $P(B)$ is the pre-probability of event B, which serves as a regularization constant and does not affect the probability results, so it can be omitted for the convenience of the calculation. Finally, the values of all the categorical variables have a post-probability through their prior probabilities and conditional probabilities, of which the highest probability is chosen as the final output forecasting value by applying one of the decision rules, such as Equation (2). $m, n$ means the number of all solar power output and weather data for calculating the conditional probability.

$$P(A_k|B_i) = arg\ max[\prod_{k=1}^{m}\prod_{i=1}^{n} P(B_i|A_k)] \tag{2}$$

In this study, prior probability means the probability that event A occurs before event B occurs, and the number of each classification variable determines the probability in advance until new data is received. Pre-probability is expressed as the ratio of the number of specific classification variables to the number of all classification variables and can be expressed as shown in Equation (3).

$$P(A_{prior}) = \frac{N(A_j)}{\sum_{k=1}^{m} N(A_k)} \tag{3}$$

$P(A_{prior})$ means the pre-probability of $A_j$, $N(A_j)$ means the number of $j$th classification variable values, and $\sum_{k=1}^{m} N(A_k)$ means the number of all classification variable values. The higher the number of data, the more variable the classification variable values are, and the pre-probability can be adjusted according to the classification variable settings, such as the decimal definition. Conditional probability means the probability of event A occurring when event B occurs, and naive in this study can be applied

as in Equation (4), because it establishes the assumption that the auxiliary variables are independent of each other.

$$P(B_i|A_k) = \prod_{i=1}^{n} P(B_i|A_k) \tag{4}$$

Methods for calculating conditional probabilities are largely divided into two types: Non-continuous and continuous. In the case of non-continuous methods, the downside is that when the meteorological figures at the time of the forecast do not exist in the classifier, the probability does not arise. For successive methods, there is an advantage that the auxiliary variable applies across almost the entire range. In this study, a probability distribution function was applied in a continuous way to respond to new values generated by natural phenomena. Conditional probabilities vary according to the function of the probability distribution defined by the user, and in this study, the most commonly used Gaussian probability distribution function was applied, as shown in Equation (5).

$$P(B_{new}|A_k) = \frac{1}{\sqrt{2\sigma_{B_iA_k}}} e^{-\frac{(B_{new} - \mu A_k B_i)^2}{2\sigma_{A_kB_i}^2}} \tag{5}$$

$(B_{new}|A_k)$ is the probability that a new auxiliary variable value will occur for a given output of solar power and is determined by the mean and variance of the values corresponding to the new auxiliary variable to be applied. $2\sigma_{B_iA_k}$ refers to the variance value of each auxiliary variable value in all classification models, and $B_{new} - \mu A_k B_i$ means the new auxiliary variable value and the average of values matching the new auxiliary variable in existing classification rules. $2\sigma_{A_kB_i}^2$ is the square value of the variance of each auxiliary variable value in all classification models.

## 2.2. SVR Model

An SVR is a regression model derived to derive a regression function from a support vector machine (SVM) used in the classification technique [29]. The SVM is used in the classification of learning data, but SVR is the method by which SVMs are normalized to predict random error values. In SVR, the linear function associated with the results is found after the data is conceived into a higher geometrical space to solve nonlinear regression problems [30]. SVR considers the following linear estimation functions [31–35]. The SVR is used to find function f(x) with minimum w in deviation from actual target $y_i$ for training data $\{x_1, y_1 \cdots x_l, y_l\} \subset x \times \mathbb{R}$ by a maximum of $\epsilon$. Where x is input vector, y is the output vector, $\mathbb{R}$ represent the input space, and linear function f(x) meeting the above condition is express as follow (6).

$$f(x) = \omega \cdot x + bias \tag{6}$$

Configure the convex optimization problem to find the minimized $\omega$, as shown in Equation (7).

$$\min \frac{1}{2}\|\omega\|^2 \qquad s.t \begin{cases} y_i - f(x) \le \epsilon \\ f(x) - y_i \le \epsilon \end{cases} \tag{7}$$

However, this equations are not established if training data are present outside the $\epsilon$-tube, as shown in Figure 2. Introduce the Slack variable $(\xi, \xi^*)$ and Cost (C) to establish the problem of convex optimization, including the data present outside the $\epsilon$-tube, and form a new problem of convex optimization, as shown in Equation (8).

$$\min \frac{1}{2}\|\omega\|^2 + C \sum_{i=1}^{n} (\zeta_i + \zeta_j^{\varphi}) \quad s.t \begin{cases} y_i - f(x) \le \epsilon + \xi_i \\ f(x) - y_i \le \epsilon + \xi_i^* \\ \xi_i, \xi_i^* \ge 0 \end{cases} \tag{8}$$

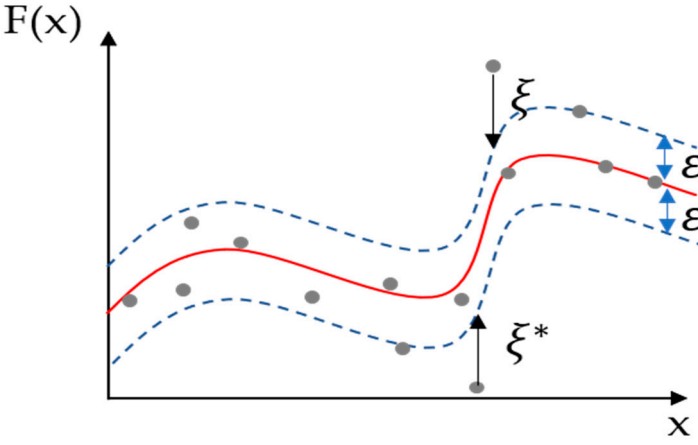

**Figure 2.** The notation of $\varepsilon$-insensitive loss function in SVR.

The optimization problem of Equation (8) is to be solved by introducing the Lagrange multiplier method where $\alpha_i, \alpha_i^*, \eta_i \eta_i^*$ is the multiplier of Lagrange parameters. For the variables in Equation (9), the solution after partial differential is obtained in Equation (10).

$$L := \tfrac{1}{2}\|\omega\|^2 + C \sum_{i=1}^{l} (\xi_i + \xi_i^*) - \sum_{i=1}^{l} (\eta_i \xi_i + \eta_i^* \xi_i^*) - \sum_{i=1}^{l} \alpha_i(\epsilon + \xi_i + y_i + \omega \cdot x_i + b)$$
$$- \sum_{i=1}^{l} \alpha_i^*\left(\epsilon + \xi_i^* + \omega \cdot x_i - b\right) \tag{9}$$

$$\therefore \omega = \sum_{i=1}^{l}(\alpha_i - \alpha_i^*)x_i, \ f(x) = \sum_{i=1}^{l}(\alpha_i - \alpha_i^*)x_i \cdot x + b \tag{10}$$

The linear SVR algorithm can be extended nonlinearly using the kernel function. Data in the input space can be approximated by using nonlinear thought function to think in dimensioned space and then approximating the nonlinear function [31–35]. The nonlinear thought function here is called the kernel function, and it is $k(x, x') \cdot \Phi(x) \cdot \Phi(x')$. Radial basis function (RBF) kernels were used as a kernel function to approximate solar power output as a nonlinear function.

*2.3. Hourly Regression Model*

Hourly regression methods are simple models that are used to identify the relationship between solar radiation and power output. During the learning period, the weight of the solar energy power is obtained for each hour and the solar power output is forecasted by utilizing the solar radiation forecast data with the input data. The mathematical model of the hourly regression model is shown in the following Equation (11).

$$P_i = a_i \times I_i \tag{11}$$

In Equation (11), $P_i$ means output for each hour, $a_i$ means weight for each hour, and $I_i$ means radiation for each hour.

**3. Forecasting Simulation of Photovoltaic Power Using Empirical Data**

In this study, input and output data for forecasting solar power output consist of one-hour data, since the unit of one hour is currently applied as a basis to operate and plan the power system of South Korea. South Korea has a temperate and cold climate and is located at the point where continental and ocean meet. The four seasons appear clearly, and the annual temperature difference is large. The test data was taken from the 'solar power plant A' located in Jeollanam-do Province. Photovoltaic power outputs data was provided by the transmission operator. The model learning period used solar radiation, temperature, humidity, and output data for one year from the time before the output

was transferred for NBC models, and model learning was performed using 720 h of solar radiation and output data for SVR models and hourly regression model. The algorithm for the forecasting model is shown in Figures 3–5. Figure 3 shows the photovoltaic forecasting algorithm using NBC. A classification rule is generated using historical data, and a probability distribution is generated using a Gaussian distribution to prepare for the case of receiving data that is not included in the classification rule. Photovoltaic power generation forecasting is performed using the conditional and prior probabilities obtained from the training data. Figure 4 shows the photovoltaic forecasting algorithm using the SVR model. The correlation analysis between past solar radiation and output is performed, and if the solar radiation is 0, the data having the output value is removed. An optimal parameter with the minimum root mean square error (RMSE) for the training period is selected and used to perform photovoltaic forecasting.

Figure 5 shows a photovoltaic forecasting model using the hourly regression model. A weight for output conversion of the solar radiation data for each training period is selected, and solar power forecasting is performed by applying the solar radiation data for the forecasting time point.

The forecasting results for 2016 were expressed using NMAE relative to the installed capacity. The NMAE is as shown in Equation (12) [36]. NMAE was calculated using 11 MW of installed capacity of photovoltaic power plant A. The forecasted sample is shown in Figure 6.

$$NMAE = \frac{1}{N}\sum_{h=1}^{N}\frac{|PP_h - PP_h^{forecast}|}{PP_{capacity}} \tag{12}$$

where $PP_h$ and $PP_h^{forecast}$ are the actual and forecasted photovoltaic power for period $h$. When the solar output is excluded from zero, and $PP_{capacity}$ refer to the installed photovoltaic power capacity.

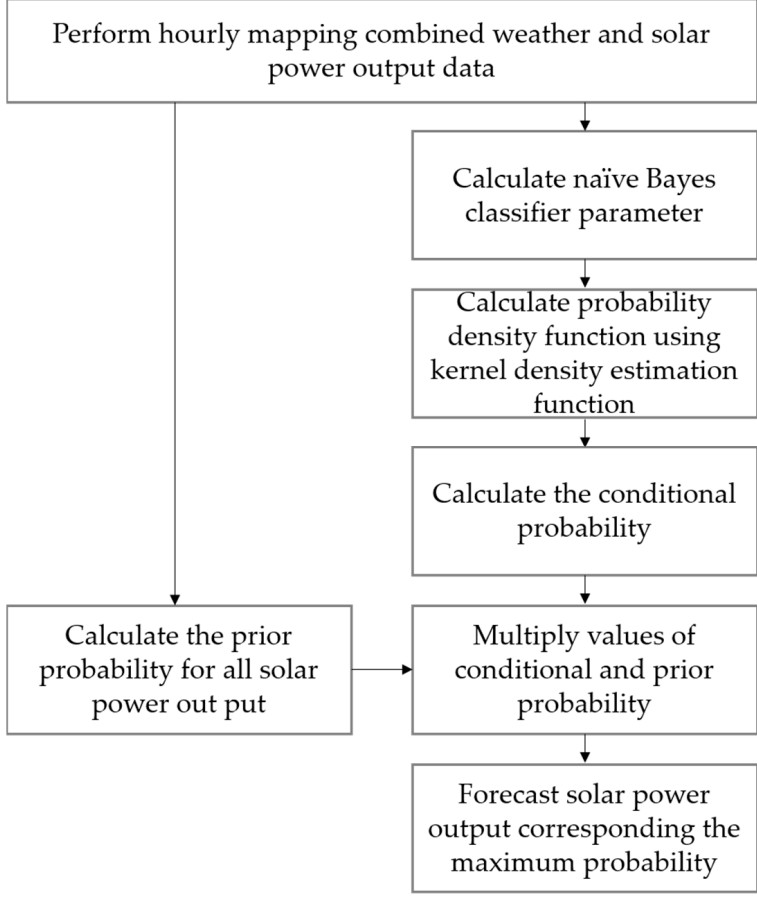

**Figure 3.** Algorithm for the NBC forecasting model.

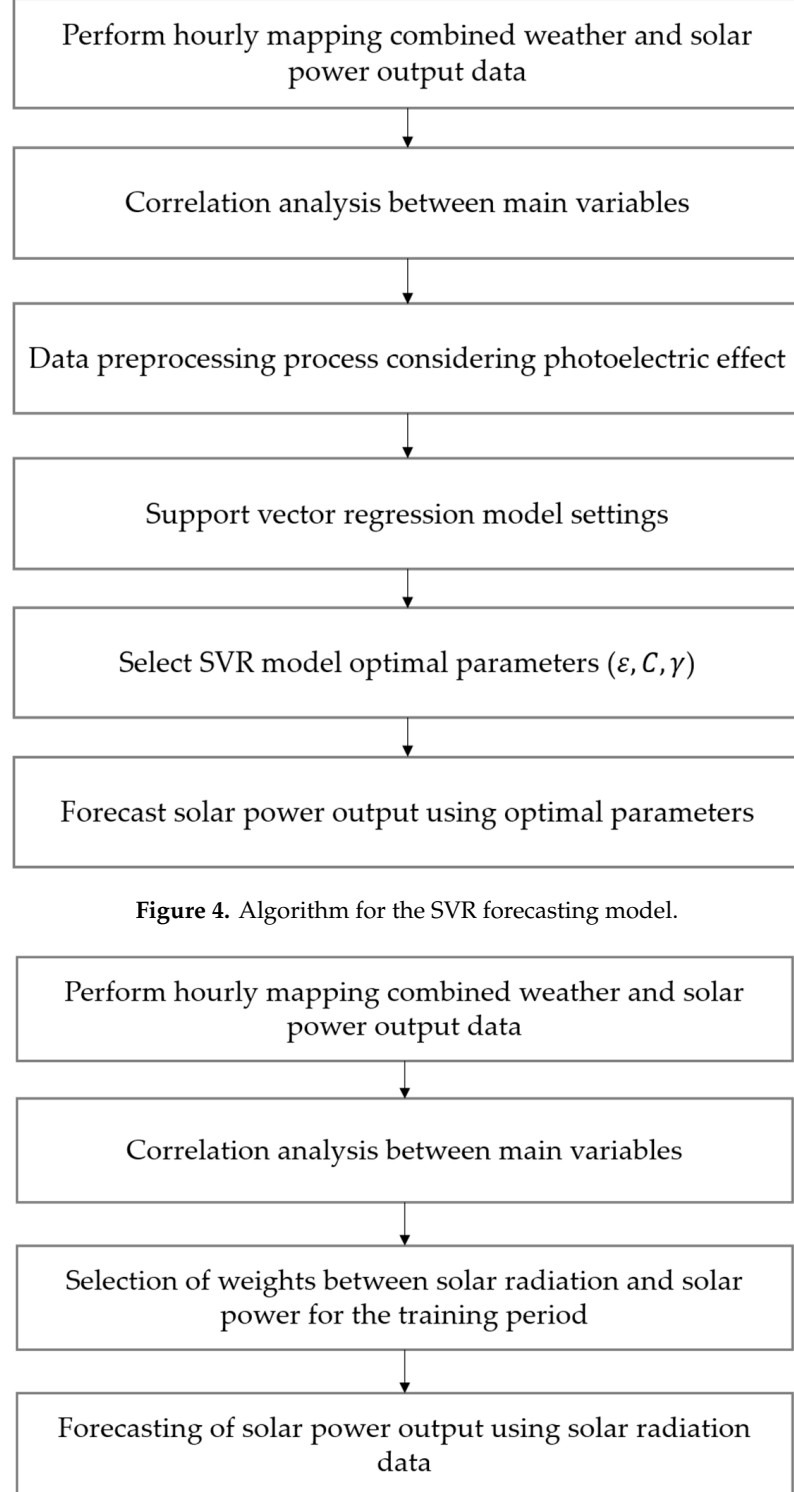

**Figure 4.** Algorithm for the SVR forecasting model.

**Figure 5.** Algorithm for the hourly regression forecasting model.

The forecast results are shown in Figures 7–9 and Table 1. The average annual forecast error for 2016 by model was 7.91% for NBC, 6.83% for SVR, and 11.75% for the year. The NBC model showed a high error in the winter, and the forecasting accuracy was high in the summer with a clear day. The SVR model had a higher forecasting accuracy than other models and showed a large forecasting error in May. The hourly regression model showed a higher prediction error than other models, and the change in forecasting performance for the season was small.

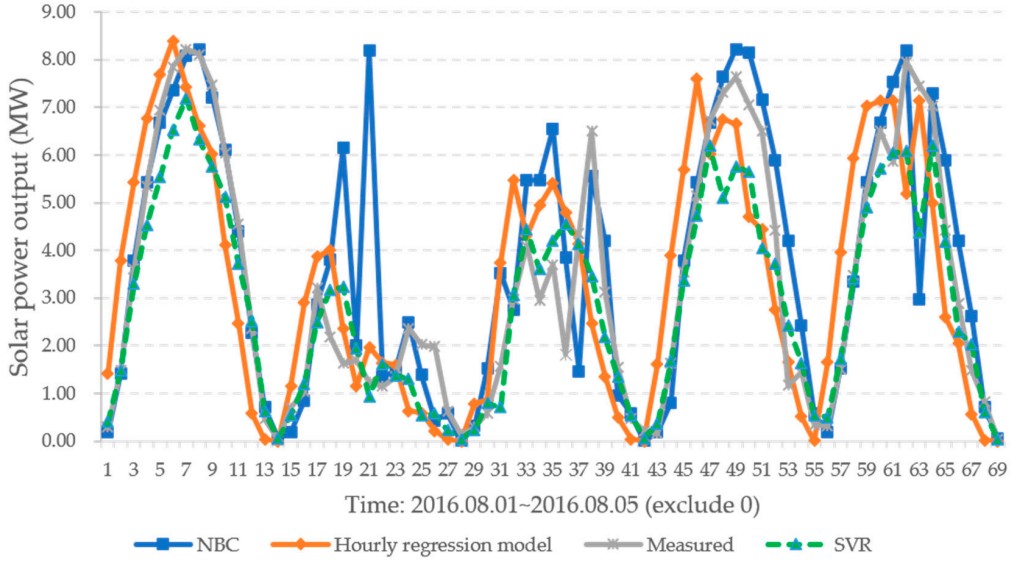

**Figure 6.** Comparison of the measured and forecast solar power.

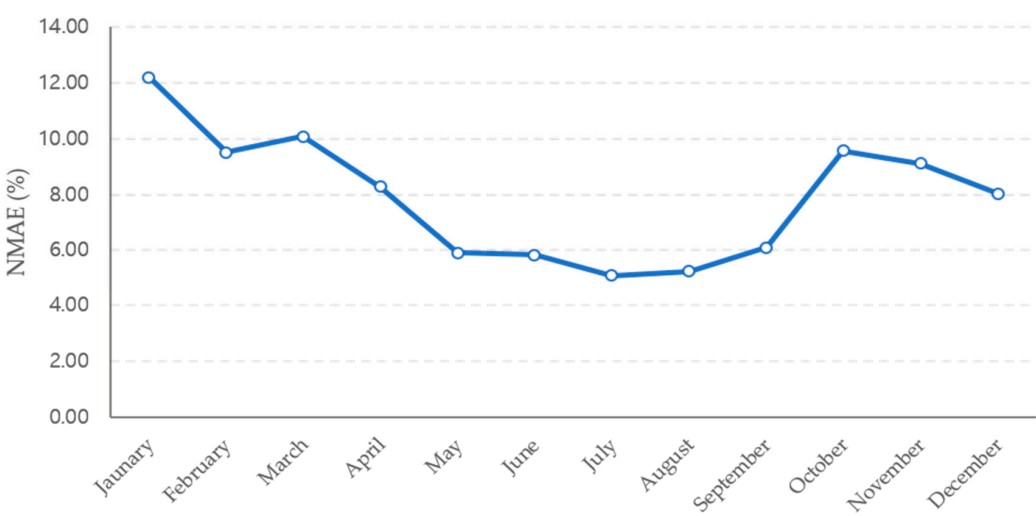

**Figure 7.** The NMAE for NBC model forecasting result in 2016.

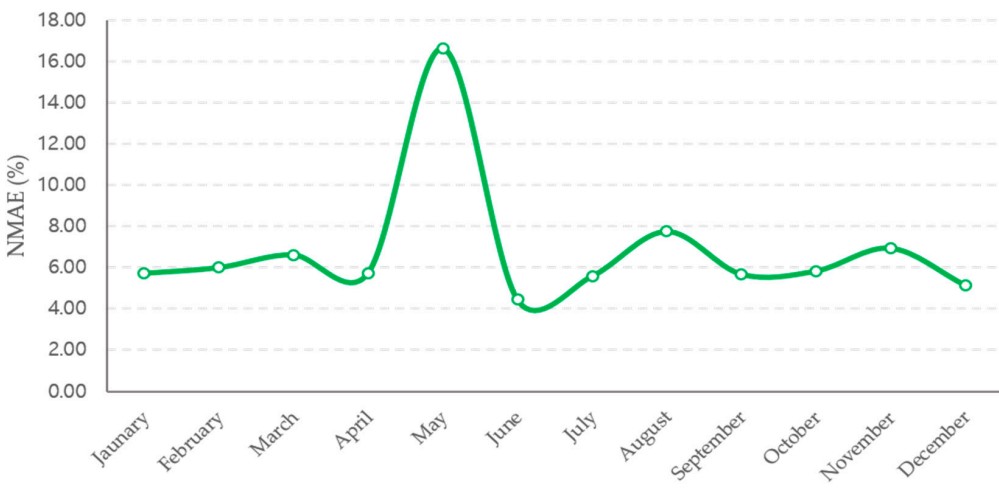

**Figure 8.** The NMAE for SVR model forecasting result in 2016.

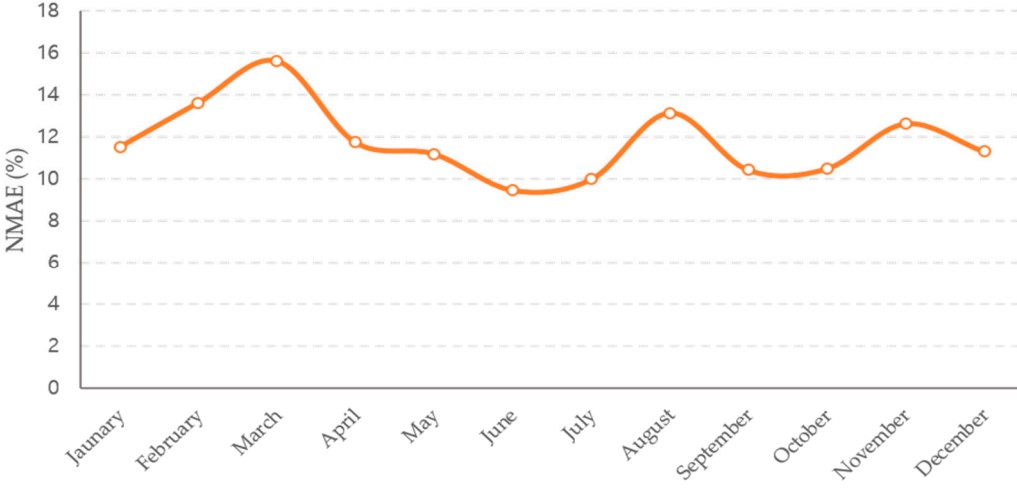

Time : 2016.01.01~2016.12.31

**Figure 9.** The NMAE for hourly regression model forecasting result in 2016.

**Table 1.** The NMAE for each model of forecasting in 2016.

| Month | NBC Model (%) | SVR Model (%) | Hourly Regression Model (%) |
|---|---|---|---|
| January | 12.19 | 5.71 | 11.52 |
| February | 9.50 | 5.99 | 13.62 |
| March | 10.08 | 6.61 | 15.60 |
| April | 8.28 | 5.73 | 11.73 |
| May | 5.90 | 16.60 | 11.18 |
| June | 5.83 | 4.43 | 9.46 |
| July | 5.08 | 5.57 | 9.98 |
| August | 6.09 | 7.74 | 13.13 |
| September | 9.57 | 5.66 | 10.41 |
| October | 9.56 | 5.82 | 10.48 |
| November | 9.10 | 6.93 | 12.61 |
| December | 8.03 | 5.13 | 11.29 |

## 4. Enhancement of Photovoltaic Power Forecasting through Ensemble

We present a forecasting upgrading technique using ensembles to improve the accuracy of solar power output forecasting and improve performance in solar power generation. The ensembles of output forecast results from NBC models, SVR models, and hourly regression models can solve the problems of overfitting that may arise from individual models and improve forecasting accuracy. To compare the ensembles method, the average output value was calculated, and the power value of the past mean absolute error (MAE) standard deviation was chosen as the weight value to compensate for the solar power output.

When the solar power output forecast was corrected through the ensemble method, the average method used had a higher error rate than SVR model with minimum error rate of 6.88% for mean and 6.52% for weight, but the accuracy of forecasting was improved. It can also reduce the large error that appears through overfitting, as shown in Figures 10–13 and Table 2. Comparisons between the SVR model with the lowest predicted error in a single model and the scatterplot between actual outputs can also be found to reduce the occurrence of large errors. The correlation between the forecasted and the measured values is the SVR model 0.9032, and the propose ensemble model 0.9399. In the propose ensemble method, the measured value and significance were increased.

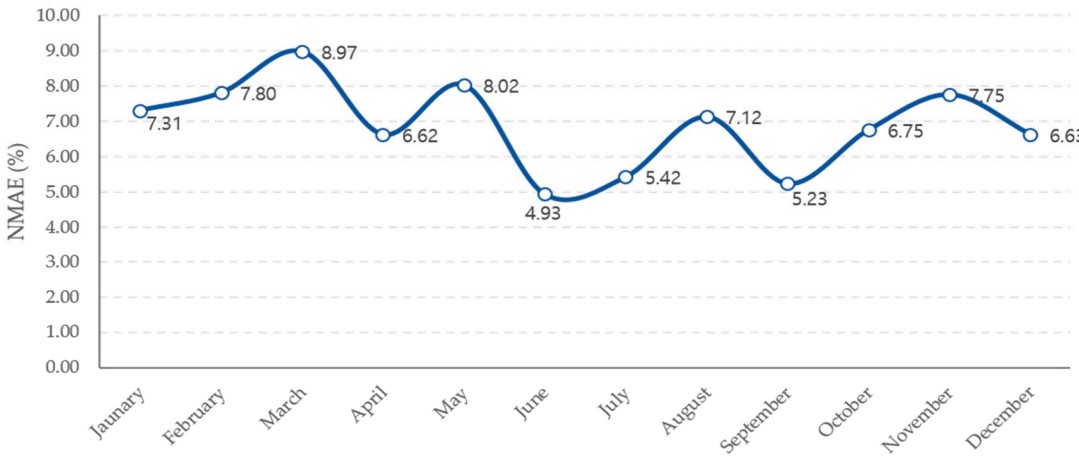

**Figure 10.** The NMAE for mean ensemble model forecasting result in 2016.

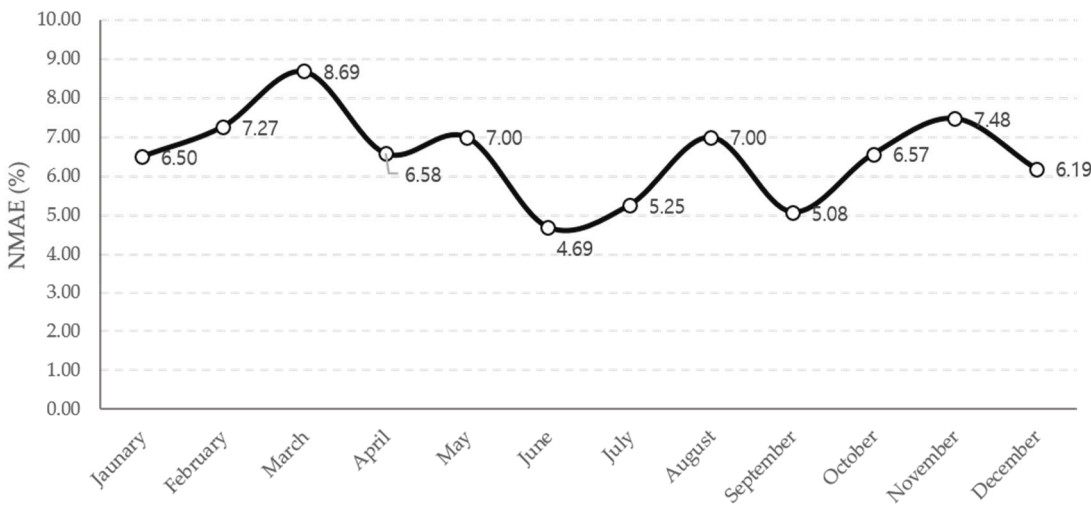

**Figure 11.** The NMAE for propose ensemble model forecasting result in 2016.

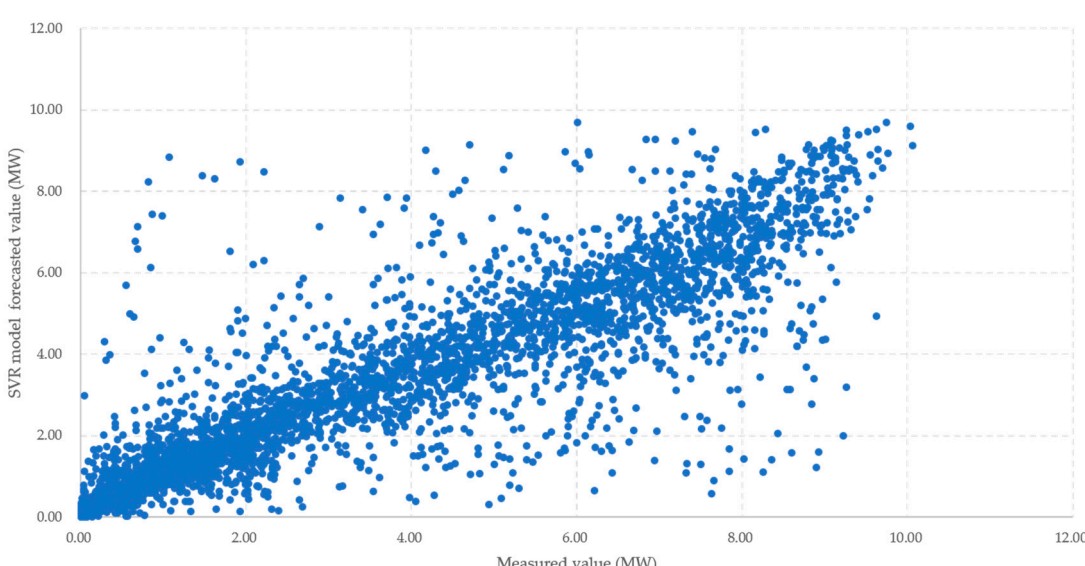

**Figure 12.** Scatterplot of measured value and SVR model forecasted value.

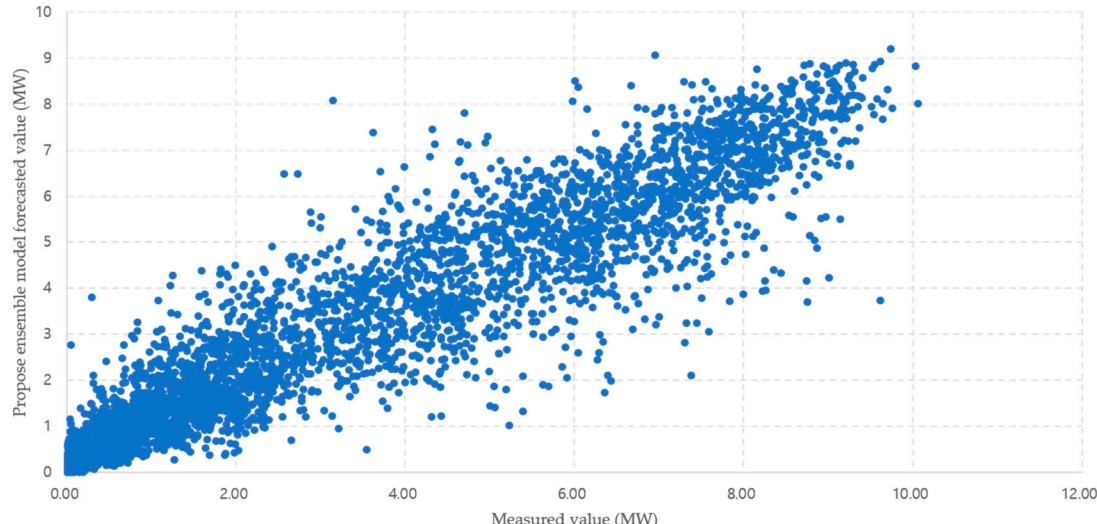

**Figure 13.** Scatterplot of measured value and propose ensemble model forecasted value.

**Table 2.** NMAE for each ensemble model of forecasting in 2016.

| Month | Mean (%) | Propose Method (%) |
|---|---|---|
| January | 7.31 | 6.50 |
| February | 8.80 | 7.27 |
| March | 9.97 | 8.69 |
| April | 6.62 | 6.58 |
| May | 8.02 | 7.00 |
| June | 4.93 | 4.69 |
| July | 5.42 | 5.25 |
| August | 7.12 | 7.00 |
| September | 5.23 | 5.08 |
| October | 6.75 | 6.57 |
| November | 7.75 | 7.48 |
| December | 6.63 | 6.19 |

## 5. Conclusions

Solar power generation is expected to have greater connectivity to the power system in response to climate change. Because power is variable by weather elements, advanced forecasting technology is required for stable operation of the power system. In this study, individual photovoltaic power forecasting was performed through NBC, SVR, and hourly regression methods. The forecast for solar power using actual data showed that the average annual average NMAE was 7.91 percent for NBC, 6.83 percent for SVR, and 11.75 percent for 2016. Ensemble techniques were introduced to improve the forecasted accuracy of solar power output, and the forecasted values were corrected by weighting the reciprocal of the standard deviation for the past error rate, resulting in an average forecasting error rate of 6.52% per year. Although single models have large errors or fail to keep up with trends in weather data, ensemble techniques have improved the accuracy of forecasts with reduced integration of predictors. In future studies, ensemble techniques with other models will be applied to improve solar power output forecasts, and studies will be conducted on how to select new weights.

**Author Contributions:** J.H. conceived and designed the overall research; K.K. implemented each forecasting model and conducted the experimental simulation; J.H. and K.K. wrote the paper; and J.H. guided the research direction and supervised the entire research process.

**Acknowledgments:** This work was supported by the Korea Institute of Energy Technology Evaluation and Planning (KETEP) and the Ministry of Trade, Industry & Energy (MOTIE) of the Republic of Korea (No. 20161210200560).

**Conflicts of Interest:** The authors declare no conflict of interest.

**Symbols**

| | |
|---|---|
| NBC | Naïve Bayes Classifier |
| NMAE | Normalized Mean Absolute Error |
| ARMA | Auto Regressive Moving Average |
| k-NN | k-Nearest Neighbors |
| NWP | Numerical Weather Prediction |
| RBF | Radial basis function |
| MAE | Mean Absolute Error |
| SVR | Support Vector Regression |
| AR | Auto-regressive |
| ANN | Artificial Neural Network |
| AI | Artificial Intelligence |
| SVM | Support Vector Machine |
| RMSE | Root Mean Square Error |

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
