# Peer review of "Weighting Factor Selection of the Ensemble Model for Improving Forecast Accuracy of Photovoltaic Generating Resources"

_energies, doi:10.3390/en12173315_

Round 1

Reviewer 1 Report

The title is too generic, please consider being more specific.

The literature review conducted is limited, please consider expanding it.

The methodology needs clarification; please consider expanding it (climatic condition, number of case studies, number of simulations…)

There are many abbreviations, please consider including a new section “Symbols”.

Author Response

Dear Editor and reviewers,

   Thank you for your comments and corrections to our revised manuscript entitled “AN ENSEMBLE MODEL FOR IMPROVING FORECAST ACCURACY OF PHOTOVOLTAIC GENERATING RESOURCES” (Manuscript ID: energies-576525). The reviewers have given many good suggestions to submit a more suitable manuscript. We tried to revise this manuscript according to these comments. The main revisions are (marked in red in the revised manuscripts) as follows. The following is the point-to-point reply to the editors.

   Thanks again for the valuable comments from the editor and reviewers.

   Yours sincerely

   Kihan Kim and Jin Hur

Reviewer 2 Report

The publication presents an interesting model for solar power forecasting.

It would be good to describe the climate for the test area where solar power data comes from.

Page 5 line 149, descriptions of Figures 3-5 should be placed in more detail in the text.

Page 6 lines 157-158, the sentences should be rewritten so that the sentence referring to formula (12) is in front of formula (12).

Page 7 figure 6 complete the data on the horizontal axis.

Page 7 line 165, figures 7-9 and results from table 1 should be further described and commented in section 3.

Pages 11-12, reformat the literature according to the editors' guidelines.

Author Response

(The authors gave the same response as above.)

Reviewer 3 Report

This paper is on developing an ensemble model for improving forecast accuracy of multiple photovoltaic power generation sources. The major information delivered is how to use the existing data to perform solar power forecasting. The following suggested changes are provided.

In Abstract, the second sentence «solar power has volatility and uncertainty due to..» may be changed into «solar power has the uncertainty due to...». In the affiliation section, the two authors from one Department, there is no need to set superscript «1» and «2». The terminology «R program» in the Abstract may be specified when the first time to use it. Why use the data «day ahead the year of 2016» may be explained because new data should be better. Check Eq.(2), what is the meanig for the italic symbol. The citation format in the Refencce section may be fixed. Check type setting in Eq. (10). On line 157, «11MW» should be «157 MW». Consider change the name of subsection title for 2.3. The original name is Physical model, however, there is no physical or experimental model presented in this section. Check the English expression throughout the manuscript.

Author Response

(The authors gave the same response as above.)

Round 2

Reviewer 1 Report

The paper has been improved.